# Health Effects of the Asthma Care Program under the Universal Coverage Scheme in Children and Young Adults in Thailand

**DOI:** 10.3390/ijerph19074130

**Published:** 2022-03-31

**Authors:** Phatthanawilai Namuenhong Inmai, Tippawan Liabsuetrakul, Nao Ichihara, Hiroyuki Yamamoto, Jutatip Thungthong, Virasakdi Chongsuvivatwong, Hiroaki Miyata

**Affiliations:** 1Health Insurance System Research Office, Health Systems Research Institute, Nonthaburi 11000, Thailand; 2Department of Epidemiology, Faculty of Medicine, Prince of Songkla University, Hat Yai 90110, Thailand; ltippawa@yahoo.com (T.L.); cvirasak@medicine.psu.ac.th (V.C.); 3Department of Healthcare Quality Assessment, Graduate School of Medicine, The University of Tokyo, Bunkyo-ku, Tokyo 113-8655, Japan; nao@g.ecc.u-tokyo.ac.jp (N.I.); yama-h@umin.ac.jp (H.Y.); h-m@keio.jp (H.M.); 4Department of Health Policy and Management, School of Medicine, Keio University, Shinjuku-ku, Tokyo 160-8582, Japan; 5National Health Security Office (NHSO), Chaengwattana Road, Lak Si 10210, Thailand; jutatip.t@nhso.go.th

**Keywords:** assessment of asthma care, asthma in children, asthma admission, universal coverage scheme, Thailand, asthma program

## Abstract

This study aimed to evaluate the effect of the asthma care program available under the Universal Coverage Scheme (UCS) in Thailand on hospital admissions per 100,000 population, its regional and seasonal variation, readmission within 28 days, and the asthma-specific fatality rate of patients aged 0–29 years in 2009–2016 compared with those in 2007–2008. A retrospective study was conducted using data sources from the UCS register and in-patient databases from the National Health Security Office (NHSO), Thailand. Hospital admissions per 100,000 population was the highest among those aged 0–4 years, but the trends decreased from 470.8 to 288.1 per 100,000 population in 2010–2014. The hospital admission rates were high in Southern Thailand and common in rainy seasons. The readmission rates within 28 days slightly decreased in all age groups in 2016 compared to those in 2007. The case fatality rate of patients aged 20–29 years decreased from 0.40% in 2007 to 0.34% in 2016. The readmission rate within 28 days and case fatality rate were the highest in patients aged 20–29 years. In conclusion, the asthma hospital admission, readmission, and case fatality rates declined over time along with the investment in the asthma care program under the UCS in Thailand. The highest hospital admission rates in patients aged 0–4 years and the readmission and case fatality rates in patients aged 20–29 years should be given more attention. Recordings of individual service utilization data in asthma patients, including quality of care provided, should be monitored to improve the asthma care system.

## 1. Introduction

Asthma is one of the most common chronic diseases that affects long-term population health. Chronic airway inflammation and narrowing of lung airways, leading to respiratory symptoms such as coughing, chest tightness, and shortness of breath, are common characteristics of asthma that require in-patient admissions for severe symptoms [1]. The data of the Global Health Data Exchange in 2017 showed that the incidence of asthma was 0.56%, whereas the prevalence and mortality rates were 3.57% and 0.006%, respectively. A high incidence was found in children aged 0–4 years, and a high prevalence was detected in children and adolescents aged 0–14 years [2]. These findings indicate that asthma is the most common chronic disease in childhood, reflecting a main cause of childhood morbidity. Improvements in asthma care and a reduction in hospital admissions are important for disease management [1]. Asthma admission rates in most European countries declined from 2001 to 2015 in children, adolescents, and adults [3]. A total of 15–50% of pediatric patients were readmitted. In the USA, the risk factors associated with asthma readmission include being of African American race, public or lack of insurance, previous hospital admission, and complex chronic comorbidity [4]. In the USA, the readmission hospitalization of children with asthma reached more than 40%. The readmission rate within 30 days totaled 2.5% [5]. Differences in the mortality of children and young adult patients are due to socioeconomic status (SES) or a life-limiting condition (LLC) [6]. The death rate of asthma patients decreased from 15 per million to 10 per million from 2001 to 2016. The highest death rate was observed in the elderly [7].

In Thailand, asthma is one of the most common non-communicable diseases, especially among younger children. A Global Asthma Network (GAN) survey in Thailand revealed that the asthma prevalence in children was over 10% [3], which supported the need for research into the hospital utilization and admission rates. The Global Initiative for Asthma (GINA) was established by the National Heart Lung and Blood Institute in collaboration with the World Health Organization in 1993. It comprises a network of individuals, organizations, and public health officials who publish information about health care patients with asthma. The GINA report, which emphasizes the Global Strategy for Asthma Management and Prevention, has been updated annually since 2002 [1]; the GINA guideline was applied in the asthma care program under the Universal Coverage Scheme (UCS) of Thailand to improve the accessibility to health care service and quality of care for asthma patients.

Initially, the asthma care program called the “Easy Asthma Clinic” was established in field-testing areas of Thailand in 2004. It covered district hospitals in which the asthma care teams consisted of general practitioners, nurses, and pharmacists. In 2009, the program was expanded to a nationwide scale involving more than 900 hospitals with 99,535 asthma patients registered with 548,583 visits [8]. In 2011, the program covered free-of-charge inhaled long-acting B2 agonists (LABAs) in combination with corticosteroids. In 2014, the program was integrated into the Rational Drug Use Hospital (RDU) project, which aimed to raise awareness among health personnel and patients [8]. Although health insurance and asthma care programs have been implemented, national monitoring was performed for the health effects on asthma patients, particularly in children and young adults who were affected the most. Therefore, this study aimed to evaluate the effects of the asthma care program available under the UCS on asthma admissions per 100,000 population, variation in regional and seasonal asthma admissions, readmissions within 28 days, and the asthma-specific fatality rate of asthma patients aged 0–4, 5–14, 15–19, and 20–29 in the periods of 2009–2011, 2011–2014, and 2014–2016, compared with the 2007–2009 period as the baseline in Thailand.

## 2. Materials and Methods

### 2.1. Study Design and Setting

A retrospective study was conducted in Thailand in January 2018–June 2019. The proposal was approved by the Ethics Research Committee of Institutional Review Board of the National Center of Global Health and Medicine, Japan, and the Institute Ethics Committee of Faculty of Medicine, Prince of Songkla University, Thailand (REC.62-025-18-1). The de-identified data were obtained under the approval of the National Health Security Office (NHSO), Thailand, which is the public organization that responds to the UCS.

### 2.2. Data Sources and Management

Two sources of databases from the NHSO, including the UCS register and in-patient databases under the UCS for asthma patients, from the fiscal year 2007 (starting in October 2007) to the fiscal year 2016 (ending in September 2016), were retrieved based on the International Classification of Diseases-10 using the principal diagnosis (pdx) of J45 and J46 (J45: asthma, J45.0: predominantly allergic asthma, J45.1: non-allergic asthma, J45.2: bronchial hyperresponsiveness, J45.8: mixed asthma, J45.9: asthma, unspecified, J46: status asthmatics). All data of asthma patients aged 0–29 years were encrypted using a personal identification number by the staff of the NHSO before providing the data to the researchers for analysis. The data from two databases were verified and cleaned for mismatched and missing data. The asthma patients were categorized into four age groups: 0–4, 5–14, 15–19, and 20–29 years. Given the different implementations of service delivery in asthma care programs in 2009, 2011, and 2014, four time periods of health effect assessment were classified as the fiscal periods of 2009–2011 (easily accessible asthma clinics in most district hospitals), 2011–2014 (free treatment of inhaled LABAs in combination with corticosteroids for asthma), and 2014–2016 (integration of asthma care with the RDU project, which aimed to raise awareness among health personnel and patients), respectively. The data during the fiscal year of 2007–2008 were considered as the baseline rate. The definition of the year period in this study was the fiscal year starting from October to September of each year.

### 2.3. Data Analysis

The data were analyzed using the STATA program version 14 (StataCorp, College Station, TX, USA). The outcome measures were asthma admissions per 100,000 population, regional and seasonal variation in hospital admission rates, readmission rates within 28 days, and case fatality rates of asthma from 2007 to 2016. Hospital admissions per 100,000 population were calculated by dividing the number of asthma admissions with the total population registered in the UCS in the same age groups and periods and then multiplying the result by 100,000. Regional variations in hospital admissions were analyzed and presented in the geographic map of Thailand, and seasonal variations were calculated in months rather than seasons because the seasons in Thailand are heterogeneous; thus, reflecting the same seasons across health regions was difficult. Readmissions within 28 days were calculated by dividing the number of readmissions within 28 days with the number of hospital admissions multiplied by 100. Case fatality rates were calculated by dividing the number of deaths from asthma in a hospital with the number of admissions multiplied by 100. The differences in asthma outcomes of patients aged 0–4, 5–14, 15–19, and 20–29 years in the different periods of 2009–2011, 2011–2014, and 2014–2016 compared with the results for the period of 2007–2009 were analyzed using one-way analysis of variance. A *p* value less than 0.05 was considered significant.

## 3. Results

### 3.1. Demographic Characteristics and Information on Asthma Hospital Admissions

The child and young adult asthma patient numbers in the UCS in Thailand were 26,397 in 2007 and 19,196 in 2016. During a 10-year study period, hospital admissions for asthma decreased by 3.16% per year, specifically decreasing from 33,661 visits to 23,030 visits from 2007 to 2016. Males showed the highest decreases of 60.3% in 2007 and 60.7% in 2016 (Table 1). The asthma hospital admissions were highest in those aged 0–4 years compared to other age groups. Moreover, patterns of asthma admission rates by age were similar among males and females.

### 3.2. The Hospital Admission per 100,000 Population in Child and Young Adult Asthma

This statistic displays the rate of asthma hospital admissions per 100,000 population in children and young adults from 2007 to 2016 in Thailand. In 2009, the asthma care program was expanded to almost all district hospitals. The GINA guideline on asthma treatment was introduced to health professionals. In 2011, the beneficiaries of the program were covered for free-of-charge inhaled LABAs in combination with corticosteroids. Then, the program was integrated with the RDU project, which aimed to raise the awareness among health personnel and patients, in 2014. Figure 1 shows the number of hospital admissions per 100,000 population in different periods of policy change. The asthma admissions per 100,000 population was the highest in individuals aged 0–4 years and decreased from 470.8 in 2010 to 288.1 per 100,000 population in 2014. Moreover, total asthma hospitalization per patient group by period differed (F = 89.72, *p* value < 0.000).

### 3.3. Regional Variation on Hospital Admission Rates in Child and Young Adult Asthma

Health regions in Thailand were divided into 13 areas using the concept of decentralization, which led to the effective and efficient management of resources. Figure 2 presents the asthma hospital admission rates of children and young adults in the 13 health regions from 2007 to 2016. The hospital admission rates in all 13 areas clearly decreased from 2007 to 2016. The asthma hospitalization of patients in all age groups varied across the health regions significantly. The findings illustrated that health region 12 had the highest hospital admission rates (10.7%), with values ranging from 9.0 to 13.5%. Health region 11 had the second highest hospital admission rates (9.8%) ranging from 8.7 to 11.4%. The lowest hospital admission rates (4.1%) in 2016 were shown in health region 2, with values in the range of 3.7 to 4.4%.

### 3.4. Seasonal Variation

Figure 3 shows the seasonal variations in asthma-related hospital admission rates by month. During the 10-year study period, the average hospitalization rate per month was 8.3%, and the standard deviation was 1.8%. The highest asthma-related hospital admission rate of about 7.6–9.7% occurred in August, from 2012 to 2016, which is the rainy season in Thailand. The lowest asthma-related hospital admission rate of 3.8–6.1% (Figure 3) occurred in April, which is the summer season. Moreover, the asthma hospitalization rates each month were different (F = 70.37, *p* value < 0.001).

Table 2 shows the readmission rate within 28 days and case fatality rate between 2007 and 2016. The readmission rate within 28 days and case fatality rate were the highest in patients aged 20–29 years. The readmission rate within 28 days slightly decreased in all age groups in 2016 compared to those in 2007. Moreover, the readmission rate within 28 days by period differed (F = 38.25, *p* value < 0.000). The case fatality rate of patients aged 20–29 years decreased from 0.40% in 2007 to 0.34% in 2016.

## 4. Discussion

The asthma care program, which was implemented at a nationwide level in Thailand, improved the accessibility to health care services and the quality of care for asthma patients under the UCS organized by the NHSO. The positive health effect of the asthma care program was not only on hospital admissions per 100,000 population but also on readmission rates and asthma-specific fatality rates over the time of the policy changes. Asthma contributed 0.4% to all hospital admissions, and the hospital admissions per 100,000 population continuously decreased. High hospital admission rates were observed during rainy seasons and in the south of Thailand. The hospital admission rate was the highest in patients aged 0–4 years, but the readmission rate within 28 days and case fatality rate were high in patients aged 20–29 years. The hospital admissions per 100,000 population in children aged 0–4 years decreased from 470.8 to 288.1 per 100,000 population between 2010 and 2014.

The findings of our study were supported by the relevant beneficiaries of asthma treatment using the GINA guidelines at district-level hospitals and well-trained asthma care teams. The medical teams consisted of general practitioners, nurses, and pharmacists. The reimbursement for LABAs in combination with corticosteroids and reimbursed expenses of medical care for hospitals from 2010 to 2012 showed a reduction in hospitalization in 2011 [9]. However, pediatric patients aged more than 5 years were controlled with a controller dose of ICS with LABAs. In addition, the use of LABAs in patients in aged 0–4 years was limited for safety. In addition, the declining trend in asthma admission rates was consistent with the information in the Global Asthma Report, which showed that the asthma admission rates of 30 European countries declined from 2001 to 2015. The asthma admission rate in Latvia was 148 per 100,000 population, decreasing by 3.12% per year. Similarly, children in Kenya showed the highest incident of asthma between 1995 and 2001. The International Study of Asthma and Allergies in Childhood showed that the prevalence of asthma in children aged 13–14 years increased from 10.4 to 13.8% in Eldoret and from 17.1 to 18.0% in Nairobi [3]. Asthma was one of main causes of hospitalization in children aged less than 5 years in our study, the same as the finding of a previous study, mostly in low- and middle-income countries (LMICs) [10]. The National Health Interview Survey data in the USA revealed that 47.2% of children under 5 years of age experienced asthma attacks in 2019 [11]. In addition, the percentage of children with asthma declined significantly from 2001 to 2016 in the United States [12]. In contrast, Dan Xu and team reported that asthma in children 6 years and older experienced more triggers than children 5 years and younger in Hangzhou. These triggers included exercise, emotional changes, house dust, pollen, renovation works at home, mosquito-repellent incense, and pets [13].

Hospital admission rates were the highest in the south of Thailand compared with those in other health regions. The rainy season there mostly occurred for 6–8 months per year, specifically from May/June to December. This season is caused by cold air, which created a relatively stable bad weather front over the southern area for several months. Considering the seasonal variation of asthma in Thailand, the greatest proportion of total asthma-related hospital admissions occurred in August, which was also part of the rainy season. This condition led to the highest hospitalization in these health regions. In the rainy season, there are various viruses and bacteria in the environment, which is reflected in an increase in viral infection. Outbreaks of cold and flu relate to high asthma hospitalizations in this season. The seasonal patterns were valuable for asthma patient management and can be used to alert patients of the increased risk in different seasons. Asthma usually presents symptoms during the rainy season. The changing weather moisture, which causes mold, is the main cause of acute asthma. This condition was associated with the Children’s Hospital of Michigan’s observation that with as little as a 10% rise in humidity, an increase in hospital visits for asthma occurred [14]. The effect of meteorological factors on hospitalization in adults revealed that temperature was the only factor associated with asthma hospitalization or emergency department visits. Temperature and relative humidity were associated with hospitalization in six studies (37%). Thunderstorms were a possible factor associated with asthma hospitalization in adults in four studies (25%) [15]. The treatment for asthma in the emergency department of the River State University Teaching Hospital, Port Harcourt, Nigeria, was more frequent in the rainy season (April–September) at 63.9%, with a peak in May [16].

In our study, the readmission rate within 28 days was 3.8–8.7%, with the highest value observed in patients aged 20–29 years, whereas the hospital admission rate was the highest in those who were aged 1–4 years. However, various factors were used to describe the high readmission rate in patients aged 20–29 years. Such a condition might have been caused by complications, co-morbid disease conditions, or other factors. Continuing research efforts should be exerted to find proof to support these assumptions. About 15–50% of pediatric patients were readmitted. The risk factors associated with asthma readmission in the USA include being of African American descent, public or lack of insurance, previous hospital admission, and complex chronic comorbidity [4]. In the USA, the hospitalization readmission rate for children with asthma was more than 40%. The readmission rate within 30 days was 2.5% [5]. From 2009 to 2013, asthma admissions in the USA totaled 1,220,047. The readmission rate within 30 days was more than 5% [17]. Among children who were hospitalized for asthma, around 20% were readmitted the next year. The readmission rate was 5.7 to 9.1% within three months. The readmission rate reduced when families received comprehensive education before discharge. In addition, the potential prevention of asthma by primary care reduced the problems [18].

The case fatality rates in this study decreased over 10 years, accounting for 0.08% in 2008 and 0.04% in 2016, respectively. The highest rates were observed in patients aged 20–29 years. The age-standardized asthma mortality rate calculated by the GAN was 3% on average among asthma patients aged from 5 to 34 years in 2015. Thailand had the fifth highest mortality rate, according to World Bank 2014, among LMICs in terms of age-standardized results for 2011–2015. The Global Asthma Report also suggested that health authorities in all countries should report the rates of asthma deaths in children to monitor the progress in asthma care and provide an early warning of epidemics of fatal asthma [3]. Similarly, the death rate for asthma decreased from 15 per million to 10 per million in 2001 to 2016. The highest death rate was observed in the elderly [7]. However, the mortality of children and young adult patients were different in terms of the SES or LLC [6]. On the other hand, aside from the asthma treatment policy resulting in hospital-related reduction in hospitalization, other effects might have occurred.

Our study supports the availability and accessibility of quality care for asthma in children and young adults as important advocacy and policy to ensure the improvement of the health and wellbeing of the population. However, several limitations were noted. First, this study was the secondary analysis of existing databases; therefore, some potential factors were not retrieved; for example, confounding effects could not be avoided. Second, the decreased hospital admissions for asthma patients may also be associated with other factors. However, the UCS covered almost all the population in Thailand, supporting the accessibility of seeking care at primary care units and outpatient care that may be related to preventable or avoidable hospitalization. Third, emergency admissions could not be extracted from the databases used. Finally, the policy of the asthma care program was implemented in only 75% of all hospitals, whereas the inpatient data of all hospitals were analyzed.

## 5. Conclusions

Asthma hospital admissions, readmission rates within 28 days, and case fatality rates have been decreasing after the implementation of the asthma care program under the UCS organized by the NHSO, Thailand. The low hospital admission rates from 2010 to 2016 in the study can be explained by the enhanced quality of health care services in the asthma care program, particularly in outpatient care. The high hospital admission rates during rainy seasons and in south of Thailand should be further explored, and appropriate strategies for proper care should be identified. The results of this evaluation will be used for planning of NHSO support in the near future. Hospital admission rates in patients aged 0–4 years and readmission and case fatality rates in patients aged 20–29 years should receive more attention. Recording of individual service utilization data in asthma patients, including the quality of care provided, must be monitored to improve the asthma care system for asthma patients.

## Figures and Tables

**Figure 1 ijerph-19-04130-f001:**
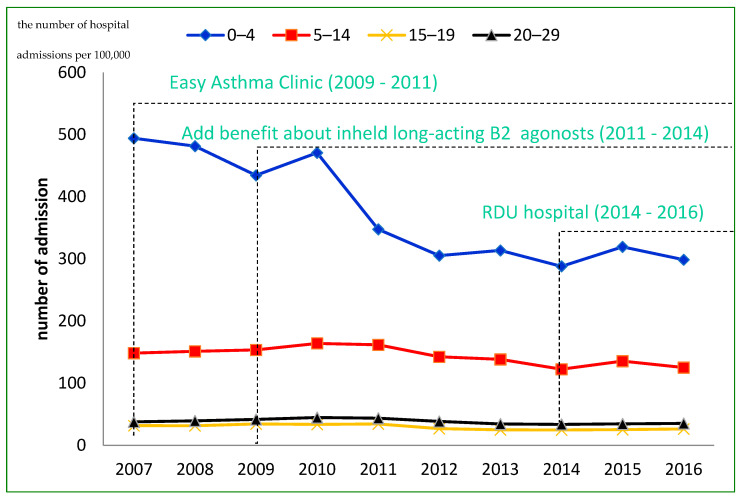
Number of hospital admissions per 100,000 population in different periods of policy change from 2007 to 2016.

**Figure 2 ijerph-19-04130-f002:**
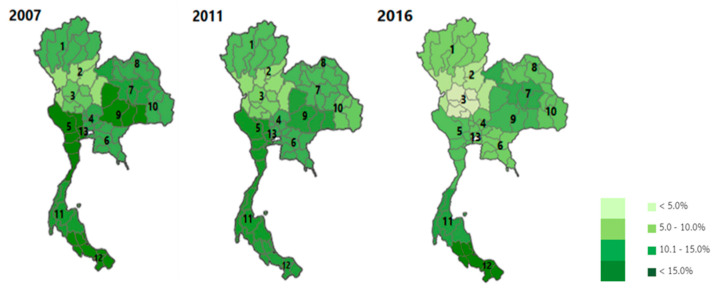
Regional variations in hospital admission rates in different periods of policy changes.

**Figure 3 ijerph-19-04130-f003:**
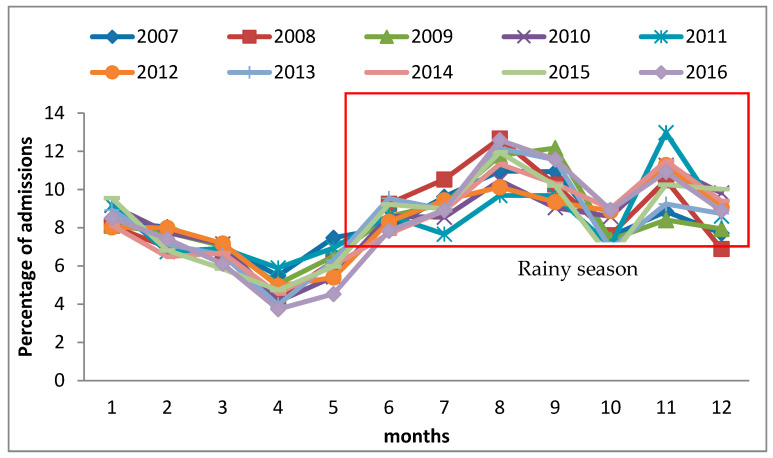
Seasonal variation in asthma-related hospital admissions rate by month.

**Table 1 ijerph-19-04130-t001:** Number and percentage of asthma admissions by age and sex distributions during the study periods of 2007–2016.

Year	Sex (%)		Age Group (%)			
	Male	Female	0–4 Years	5–14 Years	15–19 Years	20–29 Years
2007	60.3	39.7	53.1	36.9	3.7	6.3
2008	60.7	39.3	52.4	37.3	3.8	6.5
2009	61.3	38.7	49.1	39.0	4.5	7.4
2010	60.5	39.5	49.8	38.6	4.1	7.5
2011	60.3	39.7	43.7	42.8	4.9	8.6
2012	60.1	39.9	44.1	42.5	4.4	9.0
2013	60.9	39.1	46.1	41.6	4.2	8.2
2014	61.1	38.9	46.4	40.6	4.6	8.4
2015	60.9	39.1	47.4	40.6	4.2	7.8
2016	60.7	39.3	47.2	39.7	4.7	8.4

**Table 2 ijerph-19-04130-t002:** Readmission rate within 28 days and case fatality rate of asthma patients by sex and age group 2007–2016.

Year	Readmission Rate within 28 Days (%)				Case Fatality Rate (%)			
	0–4 Years	5–14 Years	15–19 Years	20–29 Years	0–4 Years	5–14 Years	15–19 Years	20–29 Years
2007	5.2	6.1	6.5	10.8	0.04	0.02	0.10	0.40
2008	5.4	5.7	7.8	8.2	0.02	0.03	0.47	0.50
2009	5.3	5.4	5.9	10.4	0.02	0.03	0.34	0.37
2010	5.0	5.2	5.5	9.6	0.01	0.02	0.17	0.50
2011	3.8	4.8	6.3	7.8	0.02	0.00	0.08	0.10
2012	4.5	4.7	5.9	7.8	0.01	0.02	0.11	0.23
2013	4.4	4.4	5.9	9.4	0.01	0.02	0.34	0.44
2014	3.9	4.1	3.8	8.2	0.04	0.00	0.56	0.67
2015	4.7	4.3	2.5	7.4	0.03	0.00	0.21	0.47
2016	4.7	4.0	3.4	9.1	0.01	0.00	0.22	0.34

## Data Availability

The data used in this study are not available in a public repository because they contain the personal identification number of the NHSO. Currently, Thailand has enforced the National Health Act to protect patient anonymity.

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
