# Peer review of "Health Effects of the Asthma Care Program under the Universal Coverage Scheme in Children and Young Adults in Thailand"

_ijerph, 2022, doi:10.3390/ijerph19074130_

Round 1

Reviewer 1 Report

Phatthanawilai Namuenhong Inmai propose a complex, interesting study about health effect of asthma care program in Thailand. This study is very well organized and it involved a lot of work. The text is sometimes difficult to understand. If it is in write author's style, I suggest that difficult expressions to be improved, for to be more attractive. 

Also, the abstract is difficut to understand in the first reading.

Author Response

Reviewer 1

Phatthanawilai Namuenhong Inmai propose a complex, interesting study about health effect of asthma care program in Thailand. This study is very well organized and it involved a lot of work.

Response: Thank you very much.

(1) The text is sometimes difficult to understand. If it is in write author's style, I suggest that difficult expressions to be improved, for to be more attractive. 

Response: The manuscript has been reviewed and revised to make more understandable.

(2) Also, the abstract is difficut to understand in the first reading.

Response: It has been revised.

 Abstract: This study aimed to evaluate the effect of asthma care program invested under the Universal Coverage Scheme (UCS) in Thailand on hospital admission per 100,000 populations, regional and seasonal variation, readmission within 28 days, and asthma-specific fatality rate of patients aged 0–29 years in 2009–2016 compared with those in 2007–2008. A retrospective study was conducted using data sources from UCS register and in-patient databases from the National Health Security Office (NHSO), Thailand. The hospital admission per 100,000 populations was the highest among those aged 0–4 years, but the trends decreased from 470.8 to 288.1 per 100,000 populations in 2010–2014. The hospital admission rates were high in southern Thailand and common in rainy seasons. The readmission rates within 28 days slightly decreased in all age groups in 2016 comparing to those in 2007. The case fatality rate of patients aged 20-29 years decreased from 0.40% in 2007 to 0.34% in 2016. The readmission rate within 28 days and case fatality rate were the highest in patients aged 20–29 years. In conclusion, the asthma hospital admission, readmission rates, and case fatality rates declined over time along with the investment in the asthma care program under the UCS in Thailand. Highest hospital admission in patients aged 0-4 years and readmission and case fatality rates in patients aged 20-29 years should be paid more attention. Recordings of individual service utilization data in asthma patients, including quality of care provided, should be monitored to improve the asthma care system.

Reviewer 2 Report

This is an interesting manuscript about the asthma care program, which was implemented at a national-wide scale in Thailand, that improved the quality of care for asthma patients.

Authors showed the positive health effect of this program on admission and on readmission  rates and asthma-specific fatality rates over time during policy changes.

Minor:

In conclusions: (main text and abstract)

Authors should add that there are the two groups of patients still require special attention - young children under 4 yo due to frequent hospital admissions and older -over 15 yo due to resubmission.

Author Response

Reviewer 2

This is an interesting manuscript about the asthma care program, which was implemented at a national-wide scale in Thailand, that improved the quality of care for asthma patients. Authors showed the positive health effect of this program on admission and on readmission  rates and asthma-specific fatality rates over time during policy changes.

Minor:

In conclusions: (main text and abstract)

(1) Authors should add that there are the two groups of patients still require special attention - young children under 4 yo due to frequent hospital admissions and older -over 15 yo due to resubmission.

Response: This has been emphasized in conclusion of abstract.

Abstract

In conclusion, the asthma hospital admission, readmission rates, and case fatality rates declined over time along with the investment in the asthma care program under the UCS in Thailand. Highest hospital admission in patients aged 0-4 years and readmission and case fatality rates in patients aged 20-29 years should be paid more attention. Recordings of individual service utilization data in asthma patients, including quality of care provided, should be monitored to improve the asthma care system.

Reviewer 3 Report

The authors present a retrospective observational epidemiological study that aims to evaluate the health effect of the Asthma Care Program Invested under the Universal Coverage Scheme in Children and Young Adults in Thailand – asthma admission per 100,000 populations, readmission within 28 days, and asthma-specific fatality rate of patients aged 0–29 years in 2009–2016 compared with those in 2007–2008 (pointed as baseline) . Additionally, analyze the chosen epidemiological indicators geographically (17 regional) and meteorologically (seasonal variation).

The title is informative and relevant. The references are relevant and recent.

The aim of the study is precise.

The study methods are well described and clear. In addition, the subject’s selection is well described.

There are some minor comments:

What is the reason for choosing these three study groups? Formally, childhood includes children 0 to 18 yrs. Are there any differences between the epidemiological  indicators in the group 15-29 yrs compared to 15-18 and 19-29?

In ICD-10, the disease codes are – J45.9, J45.9 instead of J458, J459…

The discussion is focused and informative. The study limitations are pointed out and addressed.

English language and style are acceptable.

Author Response

Reviewer 3

The authors present a retrospective observational epidemiological study that aims to evaluate the health effect of the Asthma Care Program Invested under the Universal Coverage Scheme in Children and Young Adults in Thailand – asthma admission per 100,000 populations, readmission within 28 days, and asthma-specific fatality rate of patients aged 0–29 years in 2009–2016 compared with those in 2007–2008 (pointed as baseline) . Additionally, analyze the chosen epidemiological indicators geographically (17 regional) and meteorologically (seasonal variation).

The title is informative and relevant. The references are relevant and recent. The aim of the study is precise. The study methods are well described and clear. In addition, the subject’s selection is well described.

Response: Thank you very much.

There are some minor comments:

(1) What is the reason for choosing these three study groups? Formally, childhood includes children 0 to 18 yrs. Are there any differences between the epidemiological  indicators in the group 15-29 yrs compared to 15-18 and 19-29?

Response: We agree with the reviewer and the age group of 15-29 years has been subcategorized to be 15-19 and 20-29 years depending on the WHO definition of adolescent and young adults in the objective line 83, methods lines 104-105 and 129, results line 198-199, Tables 1-2, Figure 1, discussion lines 212, 256, 258, and conclusion line 306.

(2) In ICD-10, the disease codes are – J45.9, J45.9 instead of J458, J459…

Response: ICD-10 have been revised as suggested in the methods lines 98-101.

Methods, lines 98-101

......based on the International Classification of Diseases-10 using the principal diagnosis (pdx) of J45 and J46 (J45: Asthma, J45.0: Predominantly allergic asthma, J45.1: Non-allergic asthma, J45.2: Bronchial hyper-responsiveness, J45.8: Mixed asthma, J45.9: Asthma, unspecified, J46: Status asthmatics).

(3) The discussion is focused and informative. The study limitations are pointed out and addressed. English language and style are acceptable.

Response: Thank you.

Reviewer 4 Report

Interesting topic and good study. English style needs improvement. Hospital  admission to be used instead of just admission. The differentiation of emergency admissions might  be difficult and could be considered  a study limitation .

-Line 17- hospital admission rate 

  • Line 35- please rephrase , the statement is not correct , since narrowing of the airways reflects chronic inflammation !
  • Line 43- a main cause of 
  • Line 135- 33,661 times to 23,030 ???
  • Line 143 - the clinic ?
  • Line 144- please rephrase
  • Line 204- asthma admission rates 
  • Line 254 - fatality rate due to asthma ? 
  • Line 274 - please rephrase

Author Response

Reviewer 4

(1) Interesting topic and good study. English style needs improvement. Hospital  admission to be used instead of just admission.

Response: It has been revised as suggested throughout manuscript.

(2) The differentiation of emergency admissions might be difficult and could be considered a study limitation.

Response: Limitation of emergency admission has been added in the discussion line 293-294 as suggested.

Discussion, lines 293-294

Third, the emergency admission could not be extracted from existed databases used.

(3) Line 17- hospital admission rate

Response: Done.

(4) Line 35- please rephrase, the statement is not correct, since narrowing of the airways reflects chronic inflammation

Response: It has been revised in the introduction lines 38-40.

Introduction, lines 38-40

Chronic airway inflammation and narrowing of lung airways, leading to respiratory symptoms, such as coughing, chest tightness, and shortness of breath, are common characteristics of asthma that require in-patient admissions in severe symptoms [1].

(5) Line 43- a main cause of 

Response: Done,

(6) Line 135- 33,661 times to 23,030 ???

Response: It has been revised in the results line 138.

Results, line 138

During a 10-year study period, the hospital admissions for asthma decreased by 3.16% per year, specifically decreasing from 33,661 visits to 23,030 visits in 2007 to 2016.

(7) Line 143 - the clinic ?

Response: It has been revised from "clinic" to "asthma care program".

Results line 146-147

In 2009, the asthma care program was expanded to almost all district hospitals.

(8) Line 144- please rephrase

Response: It has been rephrased in line 147-148.

Results line 147-148

The GINA guideline on asthma treatment was introduced to the health professionals.

(9) Line 204- asthma admission rates 

Response: Done in line 221.

Discussion, line 221

In addition, the declining trend of asthma admission rates was consistent with the information in the Global Asthma Report, which showed that the asthma admission rates of 30 European countries declined from 2001 to 2015.

(10) Line 254 - fatality rate due to asthma ? 

Response: It has been rephrased in line 271-272.

Discussion, line 271-272

The case fatality rates in this study decreased in 10 years accounted for 0.08% in 2008 and 0.04% in 2016, respectively.

(11) Line 274 - please rephrase

Response: It has been rephrased in line 294-295.

Discussion, line 294-295

Finally, the policy of asthma care program was implemented in only 75% of all hospitals whereas the inpatient data of all hospital were analyzed.
